# Rumen Bacterial Community of Grazing Lactating Yaks (*Poephagus grunniens*) Supplemented with Concentrate Feed and/or Rumen-Protected Lysine and Methionine

**DOI:** 10.3390/ani11082425

**Published:** 2021-08-18

**Authors:** Hu Liu, Hui Jiang, Lizhuang Hao, Xuliang Cao, Allan Degen, Jianwei Zhou, Chengfu Zhang

**Affiliations:** 1State Key Laboratory of Barley and Yak Germplasm Resources and Genetic Improvement, Institute of Animal Science and Veterinary, Tibet Academy of Agricultural and Animal Husbandry Sciences, Lhasa 850002, China; liuh2018@lzu.edu.cn (H.L.); jianghui03201119@163.com (H.J.); zhoujw@lzu.edu.cn (J.Z.); 2State Key Laboratory of Grassland Agro-Ecosystems, School of Life Sciences, Lanzhou University, Lanzhou 730000, China; caoxl19@lzu.edu.cn; 3Key Laboratory of Plateau Grazing Animal Nutrition and Feed Science of Qinghai Province, Academy of Animal Science and Veterinary Medicine of Qinghai University, Xining 810016, China; lizhuanghao1122@foxmail.com; 4Desert Animal Adaptations and Husbandry, Wyler Department of Dryland Agriculture, Blaustein Institutes for Desert Research, Ben-Gurion University of Negev, Beersheba 8410500, Israel; degen@bgu.ac.il

**Keywords:** bacterial diversity, lactating yaks, supplementation, rumen-protected amino acids, 16S rDNA sequencing

## Abstract

**Simple Summary:**

Ruminal microorganisms, especially bacteria, play a vital role in utilizing fibrous material in ruminants. The yak is a bovid on the Qinghai-Tibet Plateau that traditionally only grazes natural pasture all year. During lactation, energy intake of yaks is often well below requirements, and yaks lose body weight. Today, to mitigate body weight losses during lactation, suckling yaks are often offered supplementary feed. This study examined the effect of dietary supplements on rumen bacteria in lactating yak. The yaks were offered supplementary concentrate feed (C), rumen-protected Lys and Met (RPA), or both (RPA+C). The ratio of the relative abundance of Firmicutes to Bacteroidetes in RPA+C was greater than in the RPA group, while there was no difference between C and RPA+C. The intakes of supplements resulted in a number of alterations in the abundances of bacteria at the genus level. When supplemented with C, yaks increased the concentration of ruminal total volatile fatty acids (VFAs), acetate, and butyrate. These results demonstrate that supplementary feed: (1) alters the composition of rumen microbiota and VFAs of lactating yaks; and (2) can be used to manipulate the composition of rumen microbiota.

**Abstract:**

Traditionally, yaks graze only natural pasture all year round without supplements. Forage intake of lactating yaks is below energy and protein requirements, even in the summer, and suckling yaks lose a substantial amount of significant body weight. Today, to mitigate the loss in body weight, supplementary feed is being offered to lactating yaks. However, the effects of supplementary feed on ruminal bacterial communities in lactating yaks is unknown. In the current study, we examined the effect of supplementary feed on ruminal microbiota, using 16S rRNA sequencing, and on volatile fatty acids (VFAs). Twenty-four lactating yaks of similar body weight (218 ± 19.5 kg) and grazing natural pasture were divided randomly into four groups and received different supplements: (1) rumen-protected amino acids (RPA); (2) concentrate feed (C); (3) RPA plus C (RPA+C); and (4) no supplements (control-CON). The concentrations of total VFAs, acetate, and butyrate were greater (*p* < 0.05) when supplemented with concentrate feed (C and RPA+C) than without concentrate feed (CON and RPA). Bacteroidetes (B) and Firmicutes (F) were the dominant ruminal bacterial phyla in all groups. The ratio of relative abundance of F:B in RPA+C was greater than in the RPA group, while there was no difference between CON and RPC (interaction, *p* = 0.026). At the genus level, the relative abundances of *Absconditabacteriales_SR1, Bacteroidales-RF16-group, Bacteroidales_BS11_gut_*
*group, Prevotellaceae,* and *Rikenellaceae_RC9_gut_group* were lesser (*p* < 0.05) with supplementary concentrate feed (C and RPA+C) than without concentrate feed (CON and RPA), whereas *Butyrivibrio_2* and *Pseudobutyrivibrio* were greater (*p* < 0.05) with supplementary rumen-protected amino acids (RPA and RPA+C) than without rumen-protected amino acids (CON and C). These results demonstrate that supplementary feed: (1) alters the composition of rumen microbiota and concentrations of ruminal VFAs in lactating yaks; and (2) can be used to manipulate the composition of rumen microbiota.

## 1. Introduction

There are approximately 16 million yaks (*Poephagus grunniens*) worldwide, with 95% in China, mainly on the Qinghai-Tibetan Plateau (QTP) [1]. They are raised between 3000 and 5000 m above sea level and are well adapted to the harsh conditions of the QTP. Yaks provide meat [2] and milk [3,4] for food, dung for fuel, and wool for clothes [5], and they serve as a cultural symbol for Tibetans [6]. They are also important in maintaining stability of the alpine ecosystem.

It was reported that the meat and the milk of yak are of better quality and have higher protein content and polyunsaturated fatty acid concentrations than cattle from the lowland, for example, Holstein cattle [2,3]. However, meat and milk production in yaks is low. The reason, at least in part, is that yaks traditionally graze only natural pasture all year round without supplements [7] and, hence, are often exposed to the harsh environment in the cold season with insufficient energy and nutrient intake. As a result, yaks generally lose body weight during the harsh winter. Today, to mitigate the losses in body weight and to improve meat and milk production, supplementary feed is being offered [3].

Yaks generally calve between April and June, when forage is scarce and the nutrients are inadequate. This is a time when lactating yaks require extra energy and nutrients to provide milk for the sucking calf. Therefore, lactating yaks are typically in negative energy balance [8], and intake of amino acids, in particular lysine (Lys) and methionine (Met) [9], is insufficient while grazing with calves, even during the summer season. Feed supplementation has become an effective management option for these yaks.

Diet is an important factor that influences the rumen microbiome. Ruminal microorganisms ferment feedstuffs, especially carbohydrates (fibers and starch), that provide precursors for energy in ruminants [10]. Supplementary protein fed to pregnant heifers enhanced bacterial populations involved in hemicellulose and pectin degradation and in ammonia assimilation [11]. Abundances of the genera in Firmicutes decreased in Holstein heifers when supplemented with lysine (Lys) and methionine (Met) [12], while abundances of *Fibrobacter succinogenes, Ruminococcus albus, Ruminococcus flavefaciens,* and *Methanogen archaea* increased in beef cattle when supplemented with concentrate feed [13].

The effect of supplementary feed on rumen microbiota of grazing, lactating yaks still remains unknown. The aim of this study was to fill this gap by examining the effect of supplementary concentrate feed and rumen-protected (RP)-Lys plus RP-Met on rumen microbiota and fermentation. Although the majority of rumen-protected amino acids avoided ruminal fermentation, portions of the rumen-protected Lys and Met were released into the rumen [14,15]. In an earlier study, microbial protein synthesis increased in response to supplementary rumen-protected Lys and Met [16].

## 2. Materials and Methods

All procedures in this study were approved by the Animal Care Committee of Lanzhou University (Protocol number: LZU 201805010).

### 2.1. Study Site

This study was done during the summer from July to August 2018 at Wushaoling Yak Research Facility of Lanzhou University (102°51.7′ E, 37°12.4′ N, altitude 3154 m above sea level), located in the northeastern part of the Qinghai-Tibetan Plateau. The dominant plant species in summer were *Potentiaal fruticosa, P. viviparum, Juncus himalensis, Deschampsia cespitosa, Festuca ovina, Saussurea amara, Carex atrofusca,* and *C. moorcroftii.* The annual mean temperature was −0.1 °C with a peak in July (average: 11.3 °C).

### 2.2. Animals, Diet, Experimental Design, and Management

Twenty-four lactating yaks with similar body weight (218 ± 19.5 kg) were divided randomly into four groups (*n* = 6 per group) and received either supplementary: (1) 15 g/day RP-Lys and 5 g/day RP-Met (RPA); (2) 1.2 kg/day concentrate feed (40% corn, 20% soybean meal, and 40% highland barley meal; C); (3) RPA and C (RPA+C); or (4) no supplements (control-CON). The rumen bypass of the two rumen-protected amino acids in the present study was ≥ 800 mg/g (data from supplier). The compositions and the chemical components of the experimental diets are presented in Table 1. The different diets were offered for 32 days, after which time, samples were collected.

The lactating yaks had free access to pasture all day, and the supplementary feed was offered once daily at 18:00. The calves ran with the dams during the day and could suck milk freely. The dams were separated from the calves overnight between 18:00 in the evening and 08:00 the next day, when they were hand-milked after the calves initiated milk letdown. The milking time was about 5 min for each yak.

### 2.3. Sample Collection

On days 1, 15, and 30, three random pasture samples (100 g DM each) of mixed forage were collected according to Ren et al. [17]. In addition, 100 g of concentrate feed were collected. All samples were stored in self-sealing plastic bags at −20 °C for later analysis.

Before the yaks were released for grazing in the morning on day 32, approximately 200 mL of rumen fluid were collected from each one using a flexible oral stomach tube (Anscitech, Wuhan, China), as described by Shen et al. [18]. The tube was cleaned thoroughly between sample collections, and the first 50 mL of fluid were discarded to minimize saliva contamination. The rumen fluid was strained through 4 layers of cheesecloth, snap-frozen immediately with liquid nitrogen, and stored at −80 °C for later analysis.

### 2.4. Feed Analyses

Forage and concentrate feed samples were weighed, air dried for 48 h, and ground to pass through a 1 mm screen. Then, dry matter (DM) content of concentrate feed and forage samples was determined by drying samples for 24 h at 105 °C in a forced-air oven (AOAC 1990), and organic matter (OM; AOAC 1990) [19] was determined as loss in dry weight upon complete combustion at 600 °C for 6 h in a muffle furnace. Total N content of concentrate feed and forage was measured by the micro-Kjeldahl N method, and crude protein (CP) was estimated as 6.25 × N. Neutral detergent fiber (NDF, assayed without a heat stable amylase and expressed inclusive of residual ash) and acid detergent fiber (ADF) of feed were estimated sequentially by a fiber analyzer (Ankom 2000; ANKOM Technology, Fairport, NY, USA), following Van Soest et al. [20] and Robertson et al. [21], respectively.

### 2.5. Rumen Fermentation Variables

Rumen fluid was centrifuged at 800× *g* and 4 °C for 15 min, and the supernatant was used to determine the concentrations of volatile fatty acids (VFAs). Ruminal VFAs were analyzed by gas chromatography (SP-3420, Beifenrili Analyzer Associates, Bejing, China) with a capillary column (AT-FFAP: 30 m × 0.32 mm). The procedure was started at an initial temperature of 90 °C, increased to 120 °C at a rate of 10 °C/min, held at 120 °C for 3 min, increased from 120 °C to 180 °C at the same rate, and held for 5 min [16].

### 2.6. DNA Extraction, 16S rRNA Gene Amplification, and Sequencing

Total DNA of rumen bacteria was extracted using the E.Z.N.A ^®^ kit (Omega Bio-tek, Norcross, GA, USA), according to the protocol of the manufacturer. The concentration and the purity of the extracted DNA were determined from the 260/280 nm ratio (1.8 to 2.2) using a NanoDrop 2000 UV-vis Spectrophotometer (Thermo Scientific, Wilmington, DE, USA), and the integrity of the extracted DNA was assessed using 1% agarose gel electrophoresis. All extracted DNA samples were frozen at −80 °C for further analysis.

The PCR amplification and the bioinformatic analysis of samples were done by Shanghai Majorbio Bio-Pharm Technology Co., Ltd. (Shanghai, China). The procedures were as follows: initial denaturation at 95 °C for 3 min, followed by 27 cycles of denaturing at 95 °C for 30 s, annealing at 55 °C for 30 s and extension at 72 °C for 45 s, and single extension at 72 °C for 10 min, ending at 4 °C. The PCR mixtures contained 4 μL of 5× TransStart FastPfu buffer, 2 μL of 2.5 mM dNTPs, 0.8 μL of forward primer (5 μM), 0.8 μL of reverse primer (5 μM), 0. 4 μL of TransStart FastPfu DNA Polymerase, 10 ng of template DNA, and ddH_2_O up to 20 μL. The PCR reactions were done in triplicate. The PCR product was extracted from 2% agarose gel and purified using the AxyPrep DNA Gel Extraction Kit (Axygen Biosciences, Union City, CA, USA) according to the manufacturer’s instructions and quantified using Quantus™ Fluorometer (Promega, Madison, WI, USA).

The V3–V4 of bacterial 16S rRNA gene were amplified using primers 338F (5′-ACTC CTACGGGAGGCAGCAG-3′) and 806R (5 -GGACTACHVGGG TWT CTAAT-3′). The bacterial 16S amplification and quality filter, cluster, and analysis of 16S rRNA sequencing data followed Liu et al. [22]. After amplification, purified amplicons were pooled in equimolar and paired-end sequenced on a platform (Illumina, Miseq PE300 platform/NovaSeq PE 250, San Diego, CA, USA) according to standard protocols by Majorbio Bio-Pharm Technology Co. Ltd. (Shanghai, China). Data were analyzed using the free online Majorbio Cloud Platform (www.Majorbio.com (accessed on 28 December 2020)).

### 2.7. Statistical Analyses

Two factors were examined in the 2 × 2 factorial design study: (1) rumen-protected AA with two levels (0 and 15 g/day PR-Lys plus 5 g/day RP-Met); and (2) concentrate feed with two levels (0 and 1.2 kg/day). Data were analyzed by two-way analysis of variance using SAS 9.2 (SAS Inst. Inc., Cary, NC, USA) with the model: Y = μ + C + AA + (C × AA) + E, Y = dependent variable; μ = treatment mean value; C = effect of concentrate; AA = effect of rumen-protected amino acids; C × AA = interaction between concentrate and rumen-protected amino acids; E = residual error. When the C × AA interaction was significant, means between yaks with and without supplementary C in the same RP Lys and RP Met treatment groups were separated by *t*-test. A *p* < 0.05 was accepted as the level of significance.

## 3. Results

### 3.1. Rumen Fermentation Parameters

The ruminal concentrations of total VFAs, acetate, and butyrate were greater (*p* < 0.05) in yaks with supplemented concentrate feed than those without (Figure 1). Neither supplementary RPA nor concentrate (C) feed affected the ruminal concentration of propionate, and there was no interaction between RPA × C (*p* > 0.05).

### 3.2. Collective Sequencing Data Summary

A total of 1,269,178 raw reads were generated from the 24 rumen fluid samples. After data processing, which consisted of filtering, quality control, assembling pared-end reads, and removal of primers, chimeras, and low-confidence singletons, 1,268,641 high quality reads with a high-quality coverage of 99.5% were used. A total of 2762 operational taxonomic units (OTUs) were detected based on 97% nucleotide sequence identity analysis among reads.

In total, 2225 OTUs (Figure 2) were shared, accounting for 85.8%, 87.2%, 88.6%, and 86.7% of the total OTUs for CON, RPA, C, and RPA+C groups, respectively. The numbers of OTUs specific to CON, RPA, C, and RPA+C were 23, 14, 15, and 15, respectively.

Alpha diversity indices, including Sobs, Shannon, Simpson, Chao, and coverage, were not affected by either rumen-protected amino acids supplements, concentrate supplement, or their interaction (*p* > 0.05; Table 2). However, ACE decreased when supplemented with concentrate (*p* = 0.046).

### 3.3. Bacterial Community Composition in the Rumen Fluid

Six phyla were identified in the rumen fluid of the four treatments. Bacteroidetes (B; 61.5%) and Firmicutes (F; 29.3%) were the dominant phyla, with lesser abundances of Tenericutes (3.09%), Spirochaetes (1.96%), Patescibacteria (1.55%), and Proteobacteria (1.31%). There was no difference in the abundance of any phyla among the yak groups (*p* > 0.05; Table 3, Figure 3); however, the C and the RPA+C groups had a greater unclassified phyla (*p* = 0.043) than groups not receiving concentrate supplement. The ratio of F:B was greater in the RPA+C group than the RPA group, while the CON and the RPA groups did not differ (interaction, *p* = 0.026).

A total of 22 genera of bacteria were identified in the rumen fluid in the four treatments (Table 4, Figure 4), with *Prevotella* the most abundant (20.4%) followed by *Rikenellaceae_RC9* (9.5%) and *F082* (7.6%). The relative abundances of *Absconditabacteriales_SR1* (*p* = 0.005)*, Bacteroidales_RF16_group* (*p* = 0.006)*, Bacteroidales_BS11_gut_group* (*p* = 0.031)*, Prevotellaceae* (*p* = 0.013), and *Rikenellaceae_RC9_gut_group* (*p* = 0.017) were lesser when supplemented with concentrate feed than when not supplemented; however, the abundances of *Butyrivibrio-2* (*p* = 0.041) and *Pseudobutyrivibrio* (*p* = 0.022) were greater in yaks when supplemented with RPA than when not supplemented. There was no interaction detected.

## 4. Discussion

### 4.1. Rumen Fermentation Parameters

The increased concentration of ruminal VFAs with supplementary concentrate feed in the present study was consistent with previous studies in dairy goats [23] and Holstein-Friesian crossbred heifers [24]. This was due to the increased fermentable substrate, mainly starch, with the concentrate feed. In general, with an increase in non-fiber carbohydrate intake, there is an increase in concentration of ruminal propionate [25]. However, this did not occur in the present study, as the concentration of acetate was greater in yaks with supplementary concentrate than those without, but concentrate did not affect the concentration of propionate. The reason was probably, at least in part, due to the specific rumen microbiome in grazing yaks, which was reported to contain a greater proportion of fiborolytic bacteria than cattle [26,27]. Supplementary rumen-protected amino acids did not affect the ruminal concentration of VFAs, which is in agreement with a previous study [28].

### 4.2. Effects of Supplementary Feeds on the Bacterial Community of the Rumen Fluid

Supplementing concentrate feed and/or rumen-protected Lys and Met had no effect on the alpha diversity of the rumen bacterial community, which was in agreement with Anderson et al.’s study on steers [29]. However, yaks with supplementary concentrate feed had the lowest ACE value, which was also reported in cattle grazing tropical pasture and supplemented with concentrate [13]. In a previous study on yaks, ruminal bacterial diversity increased with an increase in forage intake because of the complexity of dietary ingredients [4]. In the present study, we reasoned that the forage intake decreased in yaks when offered supplementary concentrate, which could explain the concomitant decrease in diversity of the ruminal bacterial community.

Accounting for 87.1% of the total detected OTUs, Bacteroidetes and Firmicutes were the dominant phyla in the rumen bacterial community, which indicates that they play an important role in the bacterial ecology and the degradation of substrates in grazing, lactating yaks. Similarly, Bacteroidetes and Firmicutes constituted the majority of bacteria in cattle [30], dairy cows [31,32], growing yaks [33] and other lactating yaks [28].

The F:B ratio is influenced by a number of factors, including dietary composition, animal species and breed, and physiological stage of the animal. It was reported that the F:B ratio generally increased when the proportion of forage intake increased [34]. However, the F:B ratio in the present study (average: 0.48) was lower than in lactating yaks raised in feedlots (average: 0.55) [28]. In the present study, the F:B ratio increased with supplementary concentrate (C and RPA+C), which was in agreement with Hu et al., who reported that increasing dietary energy increased the F:B ratio [31]. Consequently, both an increase in the proportion of forage and an increase in dietary energy can increase the F:B ratio under certain conditions.

*Prevotella-1* was the most dominant genera in the phyla of Bacteroidetes, followed by *Rikenellaceae_RC9_gut_group*, which was in agreement with previous findings in yaks [26,31,35], sheep [36], and dairy cows [37]. Zhao et al. [28], however, reported that *Rikenellaceae_RC9_gut_group* was the most dominant genus in the phyla of Bacteroidetes in indoor lactating yaks, followed by *Prevotella-1* [28].

*Pseudobutyrivibrio* is a butyrate-producing, fibrolytic bacteria [38], and the C group had the lowest relative abundance of this genus. This suggested that yaks in the C group had a lower proportional intake of natural forage than the control yaks and, consequently, a lesser need of fibrolytic bacteria, which could explain this difference. Yaks receiving rumen-protected amino acids (RPA and RPA+C) had higher relative abundances of *Butyrivibrio-2* and *Pseudobutyrivibrio* than yaks without supplementary rumen-protected amino acids (CON and C), which was consistent with the greater ruminal concentrations of butyrate. *B**utyrivibrio_2* was reported to increase energy and nitrogen supplies [39].

The relative abundance of the genus *uncultured_o_Absconditabacteriales* (SR1) was lesser in yaks with supplementary concentrate feed than those without. *Absconditabacteriales* was linked with inflammatory bowel diseases [40,41] and periodontal diseases in humans [42]. In addition, *Absconditabacteriales* was associated with an unhealthy udder, and it was suggested that this genus could be used as a biomarker to assess mastitis in dairy cows [43,44]. *Bacteroidales RF16 group* was more abundant in the control yaks than the other three groups, which was in agreement with Xue et al. [45] and Liu et al. [22], who reported the greatest abundance in grazing yaks without supplements. An earlier study reported that the relative abundance of *Bacteroidales_BS_11_gut_group* was greater in yaks grazing natural pasture than yaks supplemented with concentrate feed [22]. *Bacteroidales_BS_11_gut_group* specializes in fermenting active hemicellulose monomeric sugars (e.g., xylose, fucose, mannose, and rhamnose) to short-chain fatty acids (e.g., acetate and butyrate), which are important for ruminant energy [46].

## 5. Conclusions

Supplementary rumen-protected amino acids had no effect on the diversity or the richness of the rumen bacterial community, but supplementary concentrate feed reduced the diversity. The supplements altered the relative abundance of dominant bacteria at the genus level but not at the phylum level. Supplementary concentrate feed decreased the relative abundance of *Abscoditabateriales-SR1, Bacteroidales_RF16_group, Bacteroidales_BS11_gut_group, Prevotellaceae,* and *Rikenellacae*
*RC9_*
*gut_group*, whereas supplementary rumen-protected amino acids increased the abundances of *Butyrivibrio_2* and *Pseudobutyrivibrio*. Collectively, *Prevotella* was the most abundant genus in the phylum Bacteroidetes. The present study demonstrated that supplementary feed: (1) alters the composition of rumen microbiota of lactating yaks; and (2) can be used to manipulate the composition of rumen microbiota. Data from the present study provide a basis for future research on supplementary feeds in grazing, lactating yaks on the Qinghai-Tibetan Plateau.

## Figures and Tables

**Figure 1 animals-11-02425-f001:**
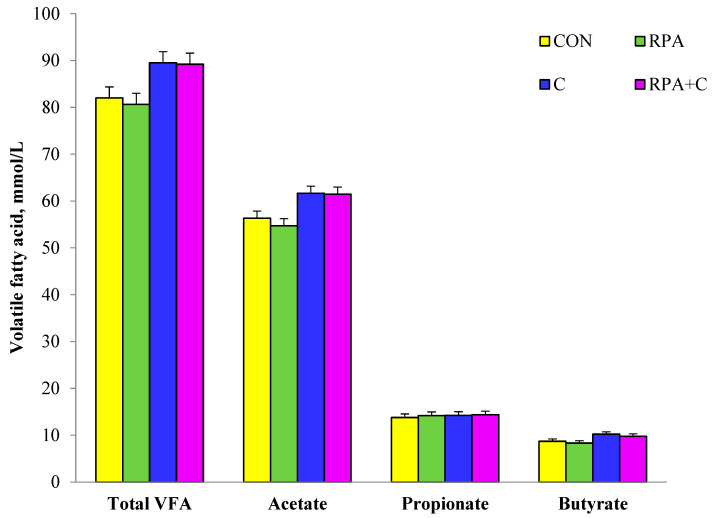
Rumen fermentation parameters of grazing, lactating yaks offered supplementary 15 g/day RP-Lys and 5 g/day RP-Met (RPA), 1.2 kg/day concentrate (C) and 15 g/day RP-Lys, 5 g/day RP-Met and 1.2 kg/day concentrate (RPA+C), or no supplement (controls-CON). Values are means ± Standard Error of Mean (SEM); *n* = 6 for each group. *p* values for: Volatile fatty acids (VFAs) for RPA, C, and RPA × C were 0.734, <0.01, and 0.828, respectively; acetate values for RPA, C, and RPA × C were 0.773, <0.01, and 0.923, respectively; propionate values for RPA, C, and RPA × C were 0.743, 0.680, and 0.823, respectively; and butyrate values for RPA, C, and RPA × C were 0.469, 0.022, and 0.957, respectively.

**Figure 2 animals-11-02425-f002:**
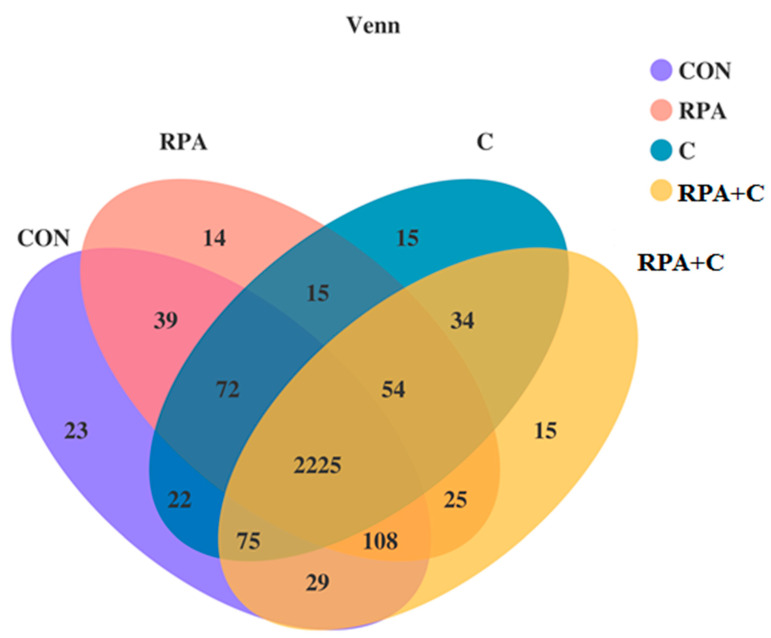
Venn diagram showing different and similar OTUs in yaks offered supplementary 15 g/day RP-Lys and 5 g/day RP-Met (RPA), 1.2 kg/day concentrate (C) and 15 g/day RP-Lys, 5 g/day RP-Met and 1.2 kg/day concentrate (RPA+C), or no supplements (control-CON).

**Figure 3 animals-11-02425-f003:**
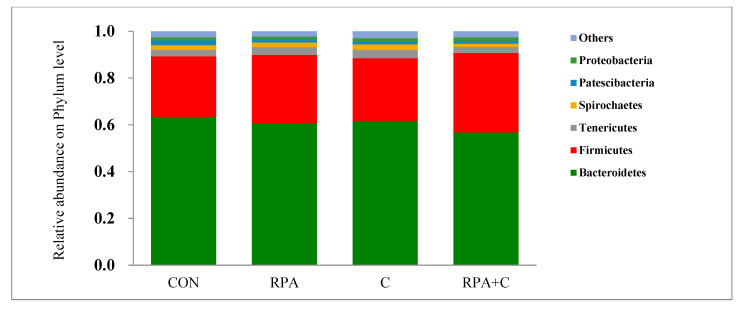
Phylum-level average relative abundances (percentage of total reads) of grazing, lactating yaks offered supplementary 15 g/day RP-Lys and 5 g/day RP-Met (RPA), 1.2 kg/day concentrate (C) and 15 g/day RP-Lys, 5 g/day RP-Met and 1.2 kg/day concentrate (RPA+C), or no supplements (control-CON).

**Figure 4 animals-11-02425-f004:**
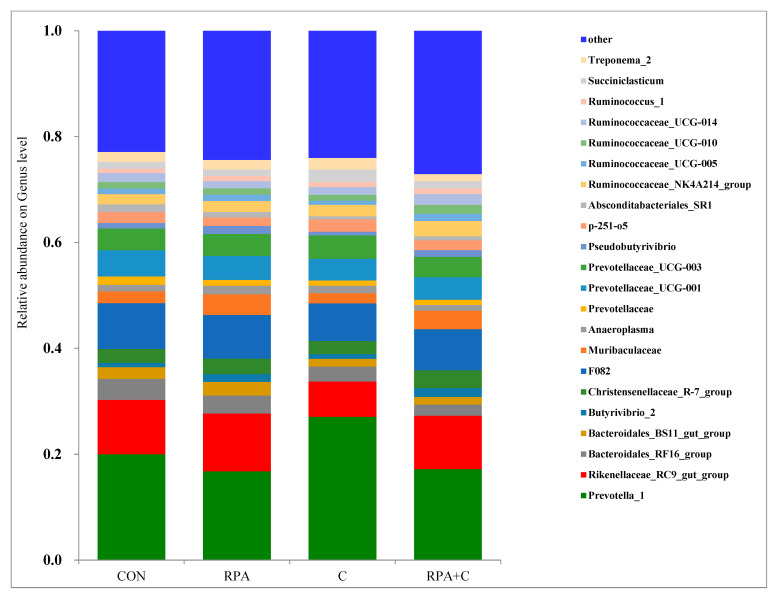
Genus-level average relative abundances (percentage of total reads) of the rumen microbial communities of grazing, lactating yaks offered supplementary 15 g/day RP-Lys and 5 g/day RP-Met (RPA), 1.2 kg/day concentrate (C) and 15 g/day RP-Lys, 5 g/day RP-Met and 1.2 kg/day concentrate (RPA+C), or no supplements (control-CON).

**Table 1 animals-11-02425-t001:** Chemical composition of the natural pasture (forage) on the rangeland and the supplementary concentrate feed (%, DM).

Items	Forage	Concentrate
DM, %	96.0	96.0
CP, %	16.7	15.3
NDF, %	50.4	10.63
ADF, %	29.9	5.52
EE, %	2.46	2.32
Ash, %	10.04	2.91
ME, MJ/kg ^1^	9.80	12.60

DM, dry matter; CP, crude protein; NDF, neutral detergent fiber; ADF, acid detergent fiber; EE, ether extract; ME, metabolizable energy. ^1^ ME was a calculated value according to Chinese Feed Ingredients and Nutritional Value Table (30th edition).

**Table 2 animals-11-02425-t002:** Effect of supplementary feeds on alpha diversity of rumen bacteria in grazing, lactating yaks.

Alpha Diversity Indices	CON	RPA	C	RPA+C	Overall Data ^1^	SEM ^2^	*p*-Values ^3^		
							RPA	C	RPA × C
Sobs	1839	1827	1713	1792	1793	46.4	0.552	0.052	0.243
Shannon	6.23	6.22	6.02	6.26	6.18	0.044	0.195	0.352	0.163
Simpson	0.0047	0.0047	0.0054	0.0045	0.0048	0.0009	0.364	0.626	0.309
ACE	2117	2106	2003	2048	2069	21.3	0.705	0.046	0.472
Chao	2141	2129	2029	2077	2094	22.5	0.694	0.079	0.486
Coverage, %	99.1	99.1	99.0	99.1	99.1	0.0003	0.984	0.604	0.270

CON, control group; RPA, with supplementary 15 g/day RP-Lys and 5 g/day RP-Met; C, with supplementary 1.2 kg/day concentrate; RPA+C, with supplementary 15 g/day RP-Lys and 5 g/d RP-Met and 1.2 kg/day concentrate. ^1^ Sequences obtained from all 24 ruminal fluid samples (average of four treatments); ^2^ SEM, standard error of mean; *n* = 6 for each group; ^3^ RPA = effect of rumen-protected amino acids; C = effect of concentrate; RPA × C = interaction between concentrate and rumen-protected amino acids.

**Table 3 animals-11-02425-t003:** Relative abundances (percentage of total reads) of rumen bacteria at phylum level in grazing, lactating yaks offered supplementary feeds.

Items	CON	RPA	C	RPA+C	Overall Data ^1^	SEM ^2^	*p*-Value ^3^		
							RPA	C	RPA × C
Bacteroidetes	64.3	61.4	62.6	57.6	61.5	2.84	0.275	0.251	0.663
Firmicutes	26.2	29.5	27.2	34.4	29.3	2.80	0.164	0.150	0.322
Tenericutes	2.82	3.33	3.60	2.59	3.09	0.476	0.612	0.973	0.144
Spirochaetes	2.05	1.98	2.36	1.43	1.96	0.411	0.254	0.775	0.318
Patescibacteria	2.03	1.53	1.31	1.34	1.55	0.231	0.333	0.078	0.281
Proteobacteria	1.36	1.07	1.38	1.41	1.31	0.177	0.573	0.176	0.228
Others	2.59	2.30	2.98	2.65	2.63	0.221	0.273	0.043	0.925
F:B ^3^ ratio	0.41	0.48	0.43	0.60	0.48	0.010	<0.001	<0.001	0.026

CON, control group; RPA, with supplementary 15 g/day RP-Lys and 5 g/day RP-Met; C, with supplementary 1.2 kg/day concentrate; RPA+C, with supplementary 15 g/day RP-Lys and 5 g/day RP-Met and 1.2 kg/day concentrate. ^1^ Sequences obtained from all 24 ruminal fluid samples (average of four treatments); ^2^ SEM, standard error of mean; *n* = 6 for each group. ^3^ RPA = effect of rumen-protected amino acids; C = effect of concentrate; RPA × C = interaction between concentrate and rumen-protected amino acids; ^3^ F:B ratio = relative abundance of Firmicutes/relative abundance of Bacteroidetes.

**Table 4 animals-11-02425-t004:** Relative abundances (percentage of total reads) of rumen bacteria at genus level in grazing, lactating yaks offered supplementary feeds.

Items	CON	RPA	C	RPA+C	Overall Data ^1^	SEM ^2^	*p*-Values ^3^		
							RPA	C	RPA × C
*Absconditabacteriales_SR1*	1.48	1.04	0.58	0.75	1.0	0.165	0.448	0.005	0.099
*Anaeroplasma*	1.25	1.62	1.42	1.07	1.3	0.341	0.964	0.594	0.315
*Bacteroidales_RF16_group*	3.96	3.42	2.87	2.09	3.1	0.399	0.169	0.006	0.728
*Bacteroidales_BS11_gut_group*	2.20	2.59	1.49	1.45	1.9	0.430	0.730	0.031	0.576
*Butyrivibrio_2*	0.83	1.46	0.86	1.68	1.2	0.310	0.041	0.685	0.774
*Christensenellaceae_R-7_group*	2.58	2.91	2.52	3.39	2.9	0.363	0.152	0.543	0.437
*F082*	8.64	8.22	7.20	7.71	7.6	0.082	0.856	0.279	0.520
*Muribaculaceae*	2.23	3.9	1.96	3.49	2.9	0.930	0.115	0.720	0.943
*Prevotella_1*	20.0	16.8	27.5	17.2	20.4	3.28	0.095	0.196	0.244
*Prevotellaceae*	1.58	1.11	1.02	1.00	1.20	0.157	0.231	0.013	0.070
*Prevotellaceae_UCG-001*	4.95	4.52	4.16	4.24	4.5	0.685	0.835	0.312	0.616
*Prevotellaceae_UCG-003*	4.10	4.16	4.50	3.85	4.2	0.575	0.672	0.913	0.452
*Pseudobutyrivibrio*	1.05	1.53	0.69	1.29	1.1	0.199	0.022	0.158	0.763
*p-251-o5*	2.05	1.59	2.37	1.85	2.0	0.304	0.220	0.202	0.902
*Rikenellaceae_RC9_gut_group*	10.3	10.9	6.79	10.1	9.5	0.86	0.065	0.017	0.108
*Ruminococcaceae_NK4A214_group*	1.91	2.06	2.23	2.89	2.3	0.399	0.419	0.086	0.419
*Ruminococcaceae_UCG-005*	1.08	1.20	0.80	1.39	1.1	0.172	0.085	0.756	0.160
*Ruminococcaceae_UCG-010*	1.20	1.22	1.09	1.68	1.3	0.191	0.171	0.317	0.124
*Ruminococcaceae_UCG-014*	1.73	1.37	1.53	2.01	1.7	0.336	0.886	0.380	0.119
*Ruminococcus_1*	0.81	0.94	0.92	1.10	0.9	0.108	0.192	0.229	0.823
*Succiniclasticum*	1.26	1.23	2.41	1.37	1.6	0.454	0.269	0.181	0.294
*Treponema_2*	1.90	1.82	2.24	1.31	1.8	0.403	0.240	0.834	0.314
*other*	22.9	24.4	24.4	27.1	24.7	1.27	0.198	0.063	0.563

CON, control group; RPA, with supplementary 15 g/day RP-Lys and 5 g/day RP-Met; C, with supplementary 1.2 kg/day concentrate; RPA+C, with supplementary 15 g/day RP-Lys and 5 g/day RP-Met and 1.2 kg/day concentrate. ^1^ Sequences obtained from all 24 ruminal fluid samples (average of four treatments); ^2^ SEM, standard error of mean; *n* = 6 for each group; ^3^ RPA = effect of rumen-protected amino acids; C = effect of concentrate; RPA × C = interaction between concentrate and rumen-protected amino acids.

## Data Availability

Data sharing is not applicable.

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
