# Peer review of "Rumen Bacterial Community of Grazing Lactating Yaks (Poephagus grunniens) Supplemented with Concentrate Feed and/or Rumen-Protected Lysine and Methionine"

_animals, 2021, doi:10.3390/ani11082425_

Round 1
Reviewer 1 Report
This is a nice and generally well written article which details supplementary feeding of yaks with additional protein to aid production. It shows some interesting information which is generally in line with what is seen in cows and sheep.
I only have a few minor comments which are detailed below.
Abstract Line 3- supplementary feeding is being offered to these yaks (reword)
Abstract- line 17- delete in
Abstract line 19- delete in the
Abstract line 19- when supplemented with rumen …. (reword)
Abstract line 20- protected is spelt incorrectly.
2nd paragraph, second line of the introduction- maybe say of Holstein cattle from the lowland?
4th line of 3rd paragraph in introduction- comma after [9]
1st line of 4th paragraph- diet is a factor, but so is age, sex, breed, temperature, other conditions etc etc
3rd sentence in 4th paragraph- in which species?
Last line of first paragraph of 2.2- comma after time
3rd line of second paragraph of 2.2- not completely sure that go on grazing makes sense. Please reword
Section 2.4- the method numbers mean very little to the readers
Section 2.6- why was a soil kit used rather than a faecal kit?
Gel electrophoresis doesn’t tell you much about DNA quality- 260/280 ratio maybe better
Line 2 of section 3.1- with, rather than without (reword)
Line 3 of section 3.1- supplementary is a typo
First line of second paragraph of section 2.2.- not completely sure that these are treatments
Paragraph 2 of section 3.3. please reword to – than when without supplementation (line 6 and 8)
Section 4.1. paragraph 1- line 4- with an increase in …. (reword)
Section 4.1. paragraph 1- line 6- present is a typo, and concentration
Section 4.1. paragraph 1- line 6- with, rather than without (reword)
Section 4.1. paragraph 1- line 9- microbiome is a typ0
Section 4.1. paragraph 1- line 9- proportion is a typo
Section 4.1. paragraph 1- line 11- ruminal is a typo
Section 4.1. paragraph 3- line 1- maybe define F :B ration
Section 4.1. paragraph 3, line 3- of doesn’t need to be in capitals
Section 4.1. paragraph 5- line 5- Butyrivibrio needs a capital
Section 4.1. paragraph 6- line 2 - with, rather than without (reword)
Section 4.1. paragraph 7- line 8 – An earlier study …. (reword)
Conclusion- paragraph – 9- demonstrated is spelt incorrectly
Table 3- please ensure that the last line aligns with the others
Author Response
Reviewer #1:
This is a nice and generally well written article which details supplementary feeding of yaks with additional protein to aid production. It shows some interesting information which is generally in line with what is seen in cows and sheep.
Response:
Thank you very much for the complimentary words.
I only have a few minor comments which are detailed below:
Abstract Line 3- supplementary feeding is being offered to these yaks (reword)
Response:
Revised, as suggested.(Abstract, line 3)
Abstract- line 17- delete in
Response:
Deleted, as suggested.
Abstract line 19- delete in the
Response:
Deleted, as suggested.
Abstract line 19- when supplemented with rumen …. (reword)
Response:
Revised, as suggested. (Abstract, line 20).
Abstract line 20- protected is spelt incorrectly.
Response:
Corrected, as suggested. (Abstract, line 21).
2nd paragraph, second line of the introduction- maybe say of Holstein cattle from the lowland?
Response:
Revised as: than cattle from the lowland, for example, Holstein cattle. (Introduction, 2nd paragraph, line 3).
4th line of 3rd paragraph in introduction- comma after [9]
Response:
Comma added, as suggested. (Introduction, 3rd paragraph, line 5).
1st line of 4th paragraph- diet is a factor, but so is age, sex, breed, temperature, other conditions etc etc
Response:
Revised, as suggested. (Introduction, 4th paragraph, line 1).
3rd sentence in 4th paragraph- in which species?
Response:
Revised as “… that provide precursors for energy in ruminants.” (Introduction, 4th paragraph, line 3).
Last line of first paragraph of 2.2- comma after time
Response:
Comma added, as suggested. (Section 2.2, line 6).
3rd line of second paragraph of 2.2- not completely sure that go on grazing makes sense. Please reword
Response:
It has been revised to: “The lactating yaks had free access to pasture all day and the supplementary feed was offered once daily at 18:00. The calves ran with the dams during the day and could suck milk freely. The dams were separated from the calves overnight between 18:00 and 08:00 (the next day), when they were hand-milked after the calves initiated milk letdown.”. (Section 2.2, 2nd paragraph, line 1 to 5).
Section 2.4- the method numbers mean very little to the readers
Response:
We deleted the method numbers.
Section 2.6- why was a soil kit used rather than a kit?
Response:
It should have been a kit. The word soil was added inadvertently and was removed in the revised version.
(Section 2.2, 1st paragraph, line 1).
Gel electrophoresis doesn’t tell you much about DNA quality- 260/280 ratio maybe better.
Response:
We revised as “The concentration and purity of the extracted DNA were determined from the 260/280 nm ratio (1.8 to 2.2) using a NanoDrop 2000 UV–vis Spectrophotometer (Thermo Scientific, Wilmington, DE, USA), and the integrity of the extracted DNA was assessed using 1% agarose gel electrophoresis.” (Section 2.6, 1st paragraph, line 2 to line 5).
Line 2 of section 3.1- with, rather than without (reword)
Response:
English is correct in original – no change made.
Line 3 of section 3.1- supplementary is a typo
Response:
Corrected, as suggested. ( Line 2 of section 3.1)
First line of second paragraph of section 3.2.- not completely sure that these are treatments
Response:
Revised as “In total, 2225 OTUs (Fig 1) were shared, accounting for 85.8%, 87.2%, 88.6% and 86.7%.” (Section 3.2, paragraph 2, line 1)
Paragraph 2 of section 3.3. please reword to – than when without supplementation (line 6 and 8)
Response:
English is correct in original – no change made.
Section 4.1. paragraph 1- line 4- with an increase in …. (reword)
Response:
Reworded, as suggested. (Section 4.1., paragraph 1- line 4)
Section 4.1. paragraph 1- line 4- present is a typo, and concentration
Response:
Corrected, as suggested. (Section 4.1. paragraph 1- line 4)
Section 4.1. paragraph 1- line 6- with, rather than without (reword)
Response:
English is correct in original – no change made.
Section 4.1. paragraph 1- line 9- microbiome is a typ0
Response:
Corrected, as suggested. (Section 4.1. paragraph 1- line 9)
Section 4.1. paragraph 1- line 9- proportion is a typo
Response:
Corrected, as suggested. (Section 4.1. paragraph 1- line 9)
Section 4.2. paragraph 1- line 11- ruminal is a typo
Response:
Corrected, as suggested.
Section 4.2. paragraph 3- line 1- maybe define F :B ration
Response:
Revised, as suggested. We defined F and B in section 3.3, paragraph 1- line 2.
Section 4.2. paragraph 3, line 3- of doesn’t need to be in capitals
Response:
It was in bold letters and this has been corrected. (Section 4.1. paragraph 3- line 3).
Section 4.2. paragraph 5- line 5- Butyrivibrio needs a capital
Response:
Revised, as suggested. (Section 4.2. paragraph 5- line 5).
Section 4.2. paragraph 6- line 2 - with, rather than without (reword)
Response:
English is correct in original – no change made.
Section 4.2. paragraph 7- line 8 – An earlier study …. (reword)
Response:
Revised, as suggested. (Section 4.2. paragraph 7- line 8).
Conclusion- paragraph – 9- demonstrated is spelt incorrectly
Response:
Corrected, as suggested. (Conclusion line 9).
Table 3- please ensure that the last line aligns with the others
Response:
Revised, as suggested.

Reviewer 2 Report
This is an interesting work dealing with an understudied species. The manuscript requires major language correction.
Although scientific sound, the manuscript has a major design problem. Authors evaluate rumen parameters testing RPA. Since the amino acid is protected it will not affect the rumen. Why did they choose to evaluate this? It makes no sense to me. The study is interesting but this is a major flaw. If published readers will not understand the rationality of using RPA and evaluate ruminal conditions.
Authors need to clarify this in the introduction and in the discussion informing the objective of such a flawed design and the rationality of using (perhaps it was part of another study dealing with milk production and other productive variables).
Title: yaks
Please add the scientific name.
The manuscript lacks a simple summary, which is mandatory.
1st and 2nd sentences of the abstract are contradictory:
Traditionally, yaks graze only natural pasture all year, without supplements. Forage intake of lactating yaks suckling their calves are below energy and protein requirements, even in the warm season. Consequently, supplementary feed is being offered these yaks.
Please rewrite. It´s not clear.
, 16S rRNA sequencing was used to examine the effect of supplementary feed on ruminal fluid volatile fatty acids
How they measured volatile fatty acids using sequencing?
You might change RPA-C to RPA+C for clarity!
The manuscript requires major language revision for clarity and to correct several mistakes and poor writing (example: were lesser (P < 0.05) in when supplemented) example: The dams were go on grazing; example: were greater (P < 0.05) in yaks with than without supplemented concentrate feed
L48: above sea level (a. s. l.)
I don´t think you need this abbreviation here
L52: tenth Panchen Lama stated that “There would be no Tibetan without 52 yaks”
I don´t think this is appropriate here. I don´t know what/who is tenth Panchen Lama. If you want to include a direct citation this is not the proper format.
Methods:
warm season
What does this mean? It´s hot obviously, but it is also wet? Give more data about the weather so people can understand the condition. The availability of grass is high in the warm season?
Authors must inform the exact time and feed condition during the rumen fluid sampling. The sampling was in the morning? Before pasture intake?
This is unusual for grass: by drying for 24 hours at 105°C
Usually, you need 72 hours at a lower temp and then you hit 105 for 24 h to DM
E.Z.N.A ® soil kit for rumen bacteria? This is previously validated?
L162: 2×2 factorial design
Since it was a 2 x 2 factorial design. Which are the factors and levels (inform here)
Results:
Figure 1
Please add information within the figure about the statistical differences.
Table 3. enlarge the table to avoid this (do it as table 4):
Ove
rall
data
Figure 2 and 3, increase the font size, it´s hard to read legend and axis.
L263: on the Sobs, Shannon, Simpson, Chao and Coverage
Poor writing, please revise.
Author Response
Reviewer #2:
This is an interesting work dealing with an understudied species.
Response:
Thank you very much for your complimentary words.
The manuscript requires major language correction. Although scientific sound, the manuscript has a major design problem. Authors evaluate rumen parameters testing RPA. Since the amino acid is protected it will not affect the rumen. Why did they choose to evaluate this? It makes no sense to me. The study is interesting but this is a major flaw. If published readers will not understand the rationality of using RPA and evaluate ruminal conditions. Authors need to clarify this in the introduction and in the discussion informing the objective of such a flawed design and the rationality of using (perhaps it was part of another study dealing with milk production and other productive variables).
Response:
The paper was edited by a native English speaker.
Thank you for your suggestion on the rumen protected AA. At the end of the introduction, we added “The rumen bypass of the two rumen protected amino acids in the present study was ≥ 800 mg/g (data from supplier); and therefore, portions of the rumen protected Lys and Met were released into the rumen [1,2]. Microbial protein synthesis increased in response to supplementary rumen protected Lys and Met supplementation [3].”
Reference:
[1] Abdelmegeid, M.K.; Elolimy, A.A.; Zhou, Z.; Lopreiato, V.; McCann, J.C.; Loor, J.J.; Rumen-protected methionine during the peripartal period in dairy cows and its effects on abundance of major species of ruminal bacteria. J Anim Sci Biotechnol. 2018, 9, 17.
[2] Fleming, A.J.; Estes, K.A.; Choi, H.; Barton, B.A.; Zimmerman, C.A.; Hanigan, M.D. Assessing bioavailability of ruminally protected methionine and lysine prototypes. J. Dairy Sci. 2019, 102, 4014-4024.
[3] Liu, H.; Yang, G.; Degen A.; Ji, K.X.; Jiao, D.; Liang, Y.P.; Xiao, L.; Long, R.J.; Zhou, J.W. Effect of feed level and supplementary rumen protected lysine and methionine on growth performance, rumen fermentation, blood metabolites and nitrogen balance in growing Tan lambs fed low protein diets. Anim. Feed Sci. Technol. 2021, 279, 115024.
Title: yaks
Please add the scientific name.
Response:
We added the scientific name “Poephagus grunniens” in the title.
The manuscript lacks a simple summary, which is mandatory.
Response:
We added a Simple Summary.
Ruminal microorganisms, especially bacteria, play a vital role in utilizing fibrous material in ruminants. The yak is a bovid on the Qinghai-Tibet Plateau that traditionally only grazes natural pasture all year. During lactation, energy intake of yaks is often well below requirements and the yaks lose body weight. Today, to mitigate body weight losses during lactation, suckling yaks are often offered supplementary feed. This study was conducted to determine the effect of dietary supplements on the rumen bacteria in lactating yak. The yaks were offered supplementary concentrate feed (C), rumen protected Lys and Met (RPA) or both (RPA+C). The ratio of the relative abundance of Firmicutes to Bacteroidetes in RPA+C was greater than in the RPA group, while there was no difference between C and RPA+C. The intakes of supplements resulted in a number of alterations in the abundances of bacteria at the genus level. When supplemented with C, yaks increased the concentration of ruminal total volatile fatty acids (VFAs), acetate and butyrate. These results demonstrate that supplementary feed: 1) alters the composition of rumen microbiota and VFAs of lactating yaks; and 2) can be used to manipulate the composition of rumen microbiota.
1st and 2nd sentences of the abstract are contradictory:
Traditionally, yaks graze only natural pasture all year, without supplements. Forage intake of lactating yaks suckling their calves are below energy and protein requirements, even in the warm season. Consequently, supplementary feed is being offered these yaks.
Please rewrite. It´s not clear.
Response:
Revised as “Traditionally, yaks graze only natural pasture all year round, without supplements. Forage intake of lactating yaks are below energy and protein requirements, even in the summer, and suckling yaks lose much body weight. Today, to mitigate the loss in body weight, supplementary feed is being offered to lactating yaks.” (Abstract, line 1 to 4).
, 16S rRNA sequencing was used to examine the effect of supplementary feed on ruminal fluid volatile fatty acids. How they measured volatile fatty acids using sequencing?
Response:
Thank you for your suggestion. We revised as “In the current study, we examined the effect of supplementary feed on ruminal microbiota, using 16S rRNA sequencing, and on volatile fatty acids (VFAs).” (Abstract, line 6 to 7).
You might change RPA-C to RPA+C for clarity!
Response:
Revised, as suggested.
The manuscript requires major language revision for clarity and to correct several mistakes and poor writing (example: were lesser (P < 0.05) in when supplemented) example: The dams were go on grazing; example: were greater (P < 0.05) in yaks with than without supplemented concentrate feed
Response:
The paper was edited by a native English speaker.
L48: above sea level (a. s. l.)
I don´t think you need this abbreviation here.
Response:
Deleted, as suggested.
L52: tenth Panchen Lama stated that “There would be no Tibetan without 52 yaks”
I don´t think this is appropriate here. I don´t know what/who is tenth Panchen Lama. If you want to include a direct citation this is not the proper format.
Response:
Deleted, as suggested.
Methods:
warm season
What does this mean? It´s hot obviously, but it is also wet? Give more data about the weather so people can understand the condition. The availability of grass is high in the warm season?
Response:
We changed the “warm season” into “summer”. We added the temperature information- The annual mean temperature was -0.1℃, with a peak in July (average: 11.3℃) in the last sentence of section 2.1.
Authors must inform the exact time and feed condition during the rumen fluid sampling. The sampling was in the morning? Before pasture intake?
Response:
Revised, as suggested. (Section 2.3, paragraph 2, line 1).
This is unusual for grass: by drying for 24 hours at 105°C
Usually, you need 72 hours at a lower temp and then you hit 105 for 24 h to DM
Response:
The samples were first air dried for 48 h. This information has been added. (Section 2.4, paragraph 1, line 1).
E.Z.N.A ® soil kit for rumen bacteria? This is previously validated?
Response:
It was a general kit. Soil was added inadvertently. The word soil has been removed in the revised version.
L162: 2×2 factorial design
Since it was a 2 x 2 factorial design. Which are the factors and levels (inform here).
Response:
We added:
Two factors were examined in the 2 × 2 factorial study: 1) rumen protected AA, with two levels (0 and 15 g/d PR-Lys plus 5 g/d RP-Met); and 2) concentrate feed with two levels (0 and 1.2 kg/d). (Section 2.7, paragraph 1, line 1 to 3).
Results:
Figure 1
Please add information within the figure about the statistical differences.
Response:
Thank you for your suggestion. This study was a 2 × 2 factorial design with no significant interactions between RPA and C. Therefore, we reasoned that it was not necessary to include the interactions.
Table 3. enlarge the table to avoid this (do it as table 4):
Ove
rall
data
Response:
Revised, as suggested.
Figure 2 and 3, increase the font size, it´s hard to read legend and axis.
Response:
Revised, as suggested.
L263: on the Sobs, Shannon, Simpson, Chao and Coverage
Poor writing, please revise.
Response:
Revised as “… had no effect on the alpha diversity of the rumen bacterial community,”.

Round 2
Reviewer 2 Report
I am satisfied with the answer and corrections provided by the authors. It´s an interesting work with an understudied species. My major concern about study design was properly answered (although, I would not choose such a design).
I have only one minor correction, at the end of the introduction change the sentence:
The rumen bypass of the two rumen-protected amino acids in the present study was ≥ 800 mg/g (data from supplier); and, therefore, portions of the rumen-protected Lys and Met were released into the rumen [14,15]. In an earlier study, microbial protein synthesis increased in response to supplementary rumen-protected Lys and Met[16].
This sentence must come before the introduction and please correct for a more general sentence because since this is your introduction, you cannot write as if it was the methods section.
Suggestion: Although the majority of rumen-protected amino acids went through ruminal fermentation intact, portions of the rumen-protected Lys and Met were released into the rumen [14,15]. In an earlier study, microbial protein synthesis increased in response to supplementary rumen-protected Lys and Met[16].
Change rumen protected to rumen-protected
Author Response
Reviewer #2:
I am satisfied with the answer and corrections provided by the authors. It´s an interesting work with an understudied species. My major concern about study design was properly answered (although, I would not choose such a design).
Response:
Thank you very much for your suggestions and complimentary words.
I have only one minor correction, at the end of the introduction change the sentence: The rumen bypass of the two rumen-protected amino acids in the present study was ≥ 800 mg/g (data from supplier); and, therefore, portions of the rumen-protected Lys and Met were released into the rumen [14,15]. In an earlier study, microbial protein synthesis increased in response to supplementary rumen-protected Lys and Met [16].
This sentence must come before the introduction and please correct for a more general sentence because since this is your introduction, you cannot write as if it was the methods section.
Suggestion: Although the majority of rumen-protected amino acids went through ruminal fermentation intact, portions of the rumen-protected Lys and Met were released into the rumen [14,15]. In an earlier study, microbial protein synthesis increased in response to supplementary rumen-protected Lys and Met [16].
Response:
Thank you for this reliable suggestion. We revised it as: “Although the majority of rumen-protected amino acids avoided ruminal fermentation, portions of the rumen-protected Lys and Met were released into the rumen [14,15]. In an earlier study, microbial protein synthesis increased in response to supplementary rumen-protected Lys and Met [16].”
Change rumen protected to rumen-protected
Response:
Revised throughout the manuscript, as suggestion.
